# Does Soil Pollution Prevention and Control Promote Corporate Sustainable Development? A Quasi-Natural Experiment of "10-Point Soil Plan" in China

**Qiong Zhou** [1], **Qian Tan** [1], **Huixiang Zeng** [1], **Yu-En Lin** [2] **and Peng Zhu** [3,*]

1   Business School, Central South University, Changsha 410083, China
2   Center for Quantitative Economics, Jilin University, Changchun 130012, China
3   School of Accounting, Hunan University of Technology and Business, Changsha 410083, China
*   Correspondence: szzhupeng@163.com; Tel.: +86-186-275-98190

**Abstract:** The Action Plan for Soil Pollution Prevention and Control ("10-point Soil Plan") provides the top-level design for soil environmental protection in China and motivates heavy polluters to participate in soil pollution prevention and control. Using a sample of Chinese-listed firms with key soil pollution regulation from 2013 to 2020, this study utilized the Difference-in-Differences method to analyze the effect and mechanism of the "10-point Soil Plan" on corporate sustainable development. The "10-point Soil Plan" significantly promoted corporate sustainability via debt vacating and cash defense effects. However, this policy failed to achieve innovation compensation. Further, the promotion of corporate sustainability via the "10-point Soil Plan" is more significant in state-owned and large enterprises and depends on the intensity of local environmental regulations. This study provides a valuable reference for government and corporates to actively implement soil pollution prevention and control measures, which complements the systematic study of soil environmental planning and helps China integrate soil environmental planning with water and air environmental planning to build a comprehensive pollution prevention system.

**Keywords:** corporate sustainable development; "10-point Soil Plan"; debt vacating effect; cash defense effect; innovation compensation effect





## 1. Introduction

"Trading the environment for growth" is a strategy adopted by developing countries to consolidate their economic foundations. This developmental model naturally leads to environmental challenges. Air and water pollution are of great concern because of their high visibility [1,2]. However, the equally serious challenge of soil pollution has yet to receive adequate attention [3,4].

Soil pollution is toxic and diffusive, which directly affects soil biological activity and changes the ecological function of soil [5,6]. Soil pollutants also easily penetrate groundwater through surface water, endangering water circulation and even the atmospheric circulation system, resulting in further deterioration of atmosphere and water [7]. These characteristics suggest that soil pollution control not only requires soil bioremediation but also addresses the pollution due to other environmental media during the remediation. Moreover, soil pollution is hard to detect or heavy because soil contamination is covert and exhibits a lag. Therefore, we need to focus on soil pollution prevention, followed by systematic soil restoration and governance.

China is one of the countries that experienced the most severe damage due to land degradation, and it has also actively explored new ways of soil environmental protection. Consequently, the Chinese government is aware of this issue. In May 2016, the State Council of China issued the Action Plan for Soil Pollution Prevention and Control ("10-point Soil Plan"), which is an environmental regulation focusing on soil pollution prevention. Among

China's numerous pollution control measures, the "10-point Soil Plan" is the first systematic regulation promulgated exclusively in the field of soil pollution. It is also the platform for soil pollution prevention and control in China during the 13th Five-Year Plan (2016–2020) and even in the longer term, which is forward-looking and epoch-making. Dynamically monitoring the effect of the "10-point Soil Plan" in China will provide valuable references for soil pollution prevention and control in other countries.

The wide range of pollution sources and the diffusion of pollutants indicate that soil pollution is a macro-topic. Soil pollution is related to sustainable development goals (SDGs). However, the prevention and control of soil pollution should be initiated by corporates via responsible action. In the absence of environmental regulations, soil pollution is becoming increasingly serious due to emissions by heavy polluters over several years, endangering their own survival and development. Unless controlled, the pollution will also threaten agricultural productivity, food safety, human health, and wellness [8,9]. After the promulgation of the "10-point Soil Plan", the Chinese government shut down more than 1300 firms involved in the heavy metal industry and implemented more than 900 projects to reduce heavy metal emissions. The development of heavy metals and other heavily polluting firms was strongly impacted. The status of corporates as the main source of soil pollution prevention and control was clarified. Firms can develop highly advanced production methods to enhance their competitiveness via pollution control. However, regulatory pressure also increases the production and operating costs of companies, which are not conducive to the sustained growth of corporate profits. Although it has been proved in the literature that environmental regulation can promote corporate sustainability [10], the promoting effect is limited by many conditions such as the types of environmental regulations [11], the intensity of environmental regulations [12], etc. Accordingly, the role of the "10-point Soil Plan" in corporate sustainability remains to be explored in depth.

Overall, most scholars are not optimistic about the "10-point Soil Plan". Hou and Li [13] point out that soil pollution prevention and control have externalities and spillover effects ranging from greenhouse gas emissions to social justice. According to Li et al. [14], the implementation of the "10-point Soil Plan" may have stimulated economic development and generated additional jobs, but also led to higher levels of air and water pollutants along with domestic supply chains. These experts are scientifically cautious about soil pollution remediation and thus doubt the sustainability of the "10-point Soil Plan" [15]. However, these studies generally focus on the impacts of the "10-point Soil Plan" at the societal level or on the physicochemical properties of contaminated soils and remediation technologies [16], and the conclusions drawn in these studies are often critical but difficult to extend to the firm level. More importantly, few studies empirically tested the policy effects of the "10-point Soil Plan", without considering the fact that firms are the main source of soil pollution prevention and control. Since the "10-point Soil Plan" is a command-and-control environmental policy, it is of great significance to focus on the end of policy transmission (firms), and it is feasible to follow the empirical articles on the "10-point Water Plan" and the "10-point Air Plan" by including firms as the research target [17,18].

Firms are the basis of soil pollution prevention and control. An in-depth analysis of corporate response to environmental policies such as the "10-point Soil Plan" can address the academic and practical concerns and improve the effectiveness of soil pollution control.

Accordingly, this study developed a Difference-in-Differences (DID) model to empirically test the micro effects of the "10-point Soil Plan" on firms and analyze the intrinsic mechanisms and heterogeneity of the effects in the context of the interactive behaviors of local governments and firms. This study found that the "10-point Soil Plan" significantly promoted corporate sustainability. However, the promotion did not originate in environmental regulation per se, but in the improved financial status of the companies due to the government's administrative intervention. Notably, the effect of innovation compensation was not realized, which may be related to the intensity of the regulation, the temporary environmental response of firms, and the low levels of soil pollution remediation technology in general. Finally, the effect of the "10-point Soil Plan" on corporate sustainability was

more significant in the case of state-owned and large enterprises, and regions with high environmental regulation intensity.

This study demonstrates that soil environmental policies can benefit firms through local government interventions, which complements the systematic study of soil environmental planning and helps China integrate soil environmental planning with water and air environmental planning to build a comprehensive pollution prevention system. The possible contributions of this study are as follows. First, this study contributes to data on the economic consequences of the "10-point Soil Plan" from the perspective of corporate sustainability. While most of the previous literature discusses the macro-level impacts of the "10-point Soil Plan" on society [13], this study places the "10-point Soil Plan" under the micro-level scenario of firms and analyzes its impact on corporate sustainability, which is complementary to the current research perspective.

Second, this study effectively identifies the specific paths of policy transmission from the central government to firms by focusing on the strategic interaction behavior of the central-local-firms in the same analytical framework. The "10-point Soil Plan" indirectly changed the financial status of companies via local government intervention. However, the expected compensation effect of innovation was not realized, indicating that the "10-point Soil Plan" relies entirely on the effectiveness of policy transmission from local governments to firms, while innovation, which is an autonomous behavior of firms, can only be stimulated through active environmental response by firms, law enforcement by local governments, and participation by other social entities.

Third, this study reveals the heterogeneity of the impact of the "10-point Soil Plan" on corporate sustainability in terms of the intensity of environmental regulations. Although the overall intensity of environmental regulation of the "10-point Soil Plan" was not adequate enough, the heterogeneity of the impact was still obvious. In areas with high environmental regulation intensity, administrative intervention by the government was more effective, and the effect of the "10-point Soil Plan" on corporate sustainability was more significant, which provides a valuable reference for decision-making to continuously promote an action plan for soil pollution prevention and control.

The next section provides the institutional background and hypothesis development. Sections 3 and 4 present the research design and results. Sections 5 and 6 analyze the mechanism and heterogeneity. The last section provides the conclusions and policy implications.

## 2. Institutional Background and Hypothesis Development

### 2.1. Institutional Background

Air, water, and soil are the three major elements in the natural environment that are related to human production and life, and they are intertwined and mutually regulated [19]. In June 2013 and February 2015, China issued the "10-point Air Plan" and "10-point Water Plan", which proposed specific action measures to improve air and water quality. However, the prevention and control of air and water pollution are obviously not adequate, because pollutants that originally belonged to air and water have the potential to migrate into the soil through natural degradation [20] and water movement [21,22]. In addition, China's rough economic development has led to high levels of total pollution emissions. Soil is the final sink for most pollutants. Therefore, the quality of the soil environment is closely linked to the goal of building a moderately prosperous society in all respects and ecological civilization. A more systematic and comprehensive strategy is needed for soil pollution control.

Therefore, on 28 May 2016, China's State Council issued the "10-point Soil Plan", which is in line with the "10-point Air Plan" and "10-point Water Plan". The "10-point Soil Plan" also uses administrative orders and controls, with the goal of controlling soil environmental risk and improving soil environmental quality in phases. The "10-point Soil Plan" is guided by three major tasks: (1) to carry out a soil pollution survey, use information technology to regularly conduct a survey and risk assessment of contaminated land, understand the soil environmental quality of agricultural land and key industrial land;

(2) to promote the prevention of soil pollution, strengthen the classification management of agricultural land based on the results of soil assessment, strengthen the access management of construction land, and strictly control the pollution of firms involved in heavy metals and mineral resources development as well as agricultural pollution; and (3) perform soil pollution control and remediation, and elucidate the responsibility of firms in soil pollution control and remediation.

The "10-point Soil Plan" is not only a useful supplement to the "10-point Air Plan" and the "10-point Water Plan", but also represents an action program for China's soil pollution prevention and control in the current and future period. The promulgation of a "10-point Soil Plan" raises China's pollution prevention and control system, which combines air, water, and soil, to a new level, and contributes to government-led, corporate-responsible, public participation and social supervision of soil pollution prevention and control and provides institutional support for achieving sustainable development goals.

### 2.2. Hypothesis Development

Theoretically, under the principal-agent model of environmental governance in China, the implementation of environmental regulation depends on the strategic interaction of the central-local-firms. The "10-point Soil Plan" is a typical command-and-control policy characterized by compulsion and timeliness. The central government clarified the responsibility of local governments in alleviating soil pollution. The administrative intervention by local governments conveyed the policy implications to firms. Accordingly, there are at least three potential mechanisms of such conscious administrative intervention by local governments to promote corporate sustainability, and their transmission paths are shown in Figure 1.

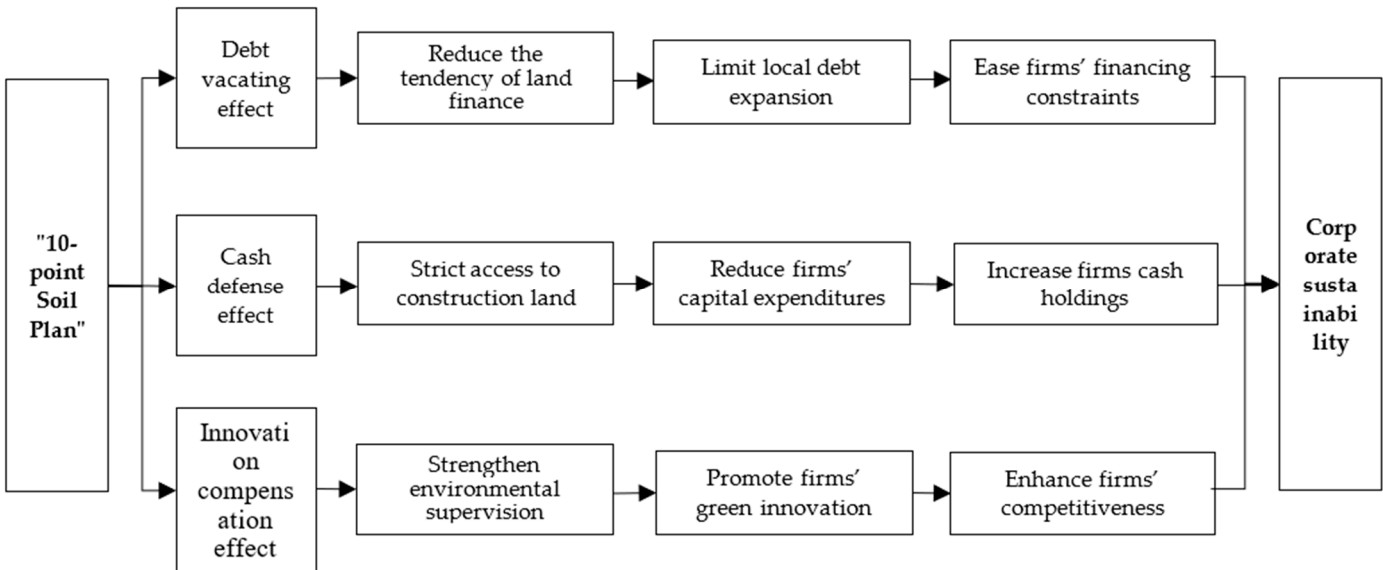

**Figure 1.** The flowchart of "10-point Soil Plan" for corporate sustainability.

### 2.2.1. Debt Vacating Effect

According to fiscal decentralization theory, the degree of fiscal decentralization in China is related to the division of interests between the central and local governments, which in turn can have a non-negligible impact on environmental governance by influencing the behavioral choices of local governments [23]. Therefore, the "10-point Soil Plan" can limit the expansion of local government debt by reducing the tendency of land finance, which in turn enhances the ability of local firms to obtain credit financing and contributes to their sustainable development. This study summarizes the above process as the local debt vacating effect. Specifically, the "10-point Soil Plan" obliges local governments to strengthen the supervision of land acquisition, transfer, and change of use, strengthen the

investigation and risk assessment of the soil environmental conditions of construction land, and implement linkage supervision of the whole process, thus improving the transparency of land transfer. As a result, local governments cannot rely on the low price of industrial land and urban development to obtain a large amount of land transfer revenue and tax revenue from land-related industries as they did before, and they cannot mortgage land for financing at will.

The land mortgage is an important mechanism of local government debt formation [24,25]. Its restriction will undoubtedly weaken the ability of the local government to raise debt. The decline of land-leasing revenue also directly affects the ability of the local government to repay debt, resulting in the local government being unable to issue new debt and expand the scale of debt for a long period of time [26,27]. Based on the crowding-out effect, the sharp expansion of local government debt will reduce credit resources in the local capital market, thereby raising the financing cost of firms and crowding out firm investment [28–30].

Therefore, when the expansion of local government debt is restricted, its debt level is lowered, which reduces the crowding out of firms' financing space by local financing platforms, alleviating the financing constraints of firms, and promoting their long-term development.

### 2.2.2. Cash Defense Effect

The "10-point Soil Plan" also increases the cash holdings of firms by reducing their capital expenditures, which can boost their ability to withstand future environmental shocks and increase their market value, which is referred to as the cash defense effect in this study. As the level of soil pollution control and remediation technology for construction land in China is still far from that of developed countries [31], the "10-point Soil Plan" does not emphasize the treatment of pollution, but rather strengthens the protection of uncontaminated soil and strictly controls new soil pollution on the construction land.

This key point of the "10-point Soil Plan" reduces the capital expenditure of firms from both subjective and objective levels. Subjectively, due to the externality of pollution, the "10-point Soil Plan" requires firms to build the main project together with the construction of soil pollution prevention and control facilities. This is contrary to the profit-seeking goal of firms, and significantly reduces the willingness of firms to invest in construction land. Objectively, local governments are required to sign a soil pollution prevention and control responsibility agreement with key industries to clarify the responsibility for pollution, which in effect increases the caution of local governments approving land for firms in key industries and hinders firms to apply for construction land.

Further, the reduction in capital expenditure is directly manifested by the increase in firms' cash holdings. Based on the agency theory of cash holdings, excessive cash holdings are prone to generate agency conflicts and damage the market value of the firms [32,33]. However, based on the trade-off theory of cash holdings, excess cash is motivated by a trade-off between the benefits and costs of holding cash, which reduces refinancing risk [34]. It increases the ability to cope with policy uncertainty [35] and allows firms to exploit profitable opportunities for future investment when this uncertainty subsides [36], then promotes future corporate performance.

In the unique context of the "10-point Soil Plan", this study argues that the excess cash holdings of heavy polluters do not cause agency problems because they are triggered by the reduction in capital expenditures and are not subjectively held by management for selfish motives. Agency problems can occur only in the event that the company's managers focus on personal interest rather than the owners' interest [37].

The excess cash has a positive effect on the firms, which can effectively enhance their market value and promote corporate sustainability comprehensively.

### 2.2.3. Innovation Compensation Effect

Porter's hypothesis argues that reasonable and strict environmental regulations are a necessary condition to ensure the win-win development of innovation-driven economic growth and environmental protection [38]. The pressure of environmental regulation can stimulate firms to embark on innovative green projects, which suggests that, although environmental regulation is uneconomic in the short run, this pressure may force firms to increase R&D investment and green innovation in the long run. The productivity and competitiveness of firms can partially or fully offset the compliance cost of environmental regulation, thus enhancing the profitability of firms, which is summarized in this study as the innovation compensation effect.

Firms engaged in key industries mentioned in the "10-point Soil Plan" face higher regulatory risks. For instance, the detection of illegal emissions, falsification of monitoring data, and other environmental violations are severely penalized and subject to special environmental enforcement. The local government is required to supervise the firms that seriously pollute the soil environment and are strongly reflected by public perception. The regulatory pressure of local governments forces firms to turn to green innovation, improve production methods, and strive to reduce pollution emissions.

However, some of the preferential measures in the "10-point Soil Plan" have enabled firms to save on the cost of compliance. The "10-point Soil Plan" repeatedly encourages firms in key industries to adopt new technologies and techniques, generating a driving force to accelerate the pace of technological innovation and change. Further, the "10-point Soil Plan" addresses the problem of green innovation requiring large amounts of capital and resource investment. First, special funds are allocated to local governments for soil pollution prevention and control. A Public–Private Partnership (PPP) model is advocated to leverage financial resources and drive social capital to participate in soil pollution prevention and control. These initiatives have partially alleviated the financial constraints and lack of resources for corporate green innovation and reduced the cost of corporate green innovation. To summarize, the "10-point Soil Plan" stimulates firms to take the lead in green innovation by assuming high regulatory risk and low compliance cost, and the savings in production cost due to green innovation can be used to meet the cost of technological conversion [39] to improve the competitiveness and thus achieve corporate sustainability [40,41].

Based on the above analysis, this study proposes the following hypothesis.

**Hypothesis 1.** *The implementation of the "10-point Soil Plan" policy can significantly promote corporate sustainability.*

### 3. Materials and Methods

This section may be divided by subheadings. It should provide a concise and precise description of the experimental results, their interpretation, as well as the experimental conclusions that can be drawn.

#### 3.1. Data Sampling

This study takes the A-share listed companies in Shanghai and Shenzhen in China from 2013 to 2020 as the initial research sample. Given that key industries are regulated by the "10-point Soil Plan", namely, non-ferrous metal mining, non-ferrous metal smelting, petroleum mining, petroleum processing, chemical industry, coking, electroplating, and tannery, the listed companies belonging to these industries were used as the experimental group in this study. Companies belonging to other industries served as the control group. The industry classification of listed companies mainly refers to the "Industry Classification Guidelines for Listed Companies (2012 Revision)" published by China Securities Regulatory Commission.

Accordingly, this study also conducted further screening of the sample as follows: (i) exclusion of the sample firms in the financial and insurance industries; (ii) exclusion

of the sample firms in the special treatment (ST), *special treatment (*ST) and particular transfer (PT) categories; (iii) and exclusion of the firms with abnormal and missing data of control variables. In addition, in order to control the effect of outliers on the results of regression, this study also adopted the 1% and 99% quantile continuous treatment of all continuous variables. Following the above screening, a total of 18,687 firm-year observations were obtained in this study.

Except for the number of green patents, which was determined from the Chinese Research Data Service Platform (CNRDS), all the other data related to firm characteristics analyzed in this study were obtained from the China Stock Market & Accounting Research Database (CSMAR) and the China Center for Economic Research (CCER). The two main sources of data at the regional level include (1) the regional marketization index obtained from the "China Marketization Index Report by Province" (2021) [42], with missing years filled in by manual calculation; and (2) the land transaction fee obtained from the "China Land and Resources Statistical Yearbook".

### 3.2. Variable

#### 3.2.1. Dependent Variable

The dependent variable in this study was corporate sustainability, which is defined as consistent profitability and leading competitiveness in an uncertain economic environment. Currently, several methods are used to measure corporate sustainability. Considering the objectivity of indicators and the availability of data, this study was based on Ain et al. [43] and Chen et al. [44] to measure corporate sustainability by calculating sustainable growth rate using Van Horn's SGM model. The sustainable growth rate (SGR) evaluates the influence of shareholders and creditors based on general financial performance indicators and reflects the sustainability of firms comprehensively. SGR is calculated as follows.

$$SGR = \frac{ROE \times}{1 - ROE \times b} \tag{1}$$

where *ROE* is the firm's return on equity and *b* denotes the firm's retained earnings ratio. The larger the calculated *SGR* value, the more sustainable is the firm.

#### 3.2.2. Independent Variable

The explanatory variable in this study was the implementation of the "10-point Soil Plan" policy, which was portrayed by the DID term (Treat × Post). Specifically, Treat was taken as 1 when the firm belonged to the industries regulated by the "10-point Soil Plan", and 0 otherwise. Considering that the "10-point Soil Plan" policy was promulgated and implemented in 2016, this study assumed 2016 as the implementation point of the policy, Post as 1 in 2016 and subsequent years; otherwise, it was assumed 0.

#### 3.2.3. Control Variables

Drawing on the previous studies, this study selected a series of indicators that affect corporate sustainability as control variables [45–48], including firm size (Size), leverage (Lev), firm age (Age), nature of ownership (Soe), ownership concentration (First), board independence (ID), and regional marketability index (MKT). The definitions and measures of all variables are summarized in Table A1.

### 3.3. Model

To test the impact of the "10-point Soil Plan" on corporate sustainability, the following DID model is constructed.

$$SGR_{it} = \alpha + \beta * Treat_i \times Post_t + \varphi X_{it} + \gamma_t + \gamma_i + \gamma_{city} + \varepsilon_{it} \tag{2}$$

The coefficient $\beta$ of the DID term (Treat × Post) is the main coefficient of interest, which reflects the net impact of the "10-point Soil Plan" on the sustainability of firms in key

regulated industries; $X_{it}$ represents the control variable; $\gamma_t$, $\gamma_i$, and $\gamma_{city}$ are the time, firm, and city fixed effects, respectively; $\varepsilon_{it}$ denotes the random disturbance term.

## 4. Results

### 4.1. Descriptive Statistics and Correlation Analysis

Based on the results of descriptive statistics (see Table 1), the mean value of sustainable growth rate (SGR) was 0.041, i.e., the average sustainable growth rate of firms was 4.1%. However, the extreme value of SGR shows that the sustainability of the sample firms varies greatly. The mean value of Treat was 0.313, which determined whether the firms were involved in key regulated industries in the sample period, indicating that 31.3% of the sample firms belonged to key regulated industries. In addition, the variance and the difference between the mean and median of each variable, such as Size and Lev, were small, indicating the relative stability of these indicators.

**Table 1.** Descriptive statistics.

| Variables | N | Mean | Std | Min | Median | Max |
|---|---|---|---|---|---|---|
| SGR | 18 687 | 0.041 | 0.105 | −0.485 | 0.046 | 0.323 |
| Treat | 18 687 | 0.313 | 0.464 | 0 | 0 | 1 |
| Post | 18 687 | 0.689 | 0.463 | 0 | 1 | 1 |
| Treat × Post | 18 687 | 0.212 | 0.409 | 0 | 0 | 1 |
| Size | 18 687 | 22.360 | 1.285 | 19.980 | 22.180 | 26.300 |
| Lev | 18 687 | 0.432 | 0.202 | 0.059 | 0.425 | 0.891 |
| Age | 18 687 | 2.877 | 0.315 | 1.946 | 2.944 | 3.466 |
| Soe | 18 687 | 0.378 | 0.485 | 0 | 0 | 1 |
| First | 18 687 | 0.341 | 0.147 | 0.085 | 0.319 | 0.737 |
| ID | 18 687 | 0.376 | 0.053 | 0.333 | 0.364 | 0.571 |
| MKT | 18 687 | 12.930 | 1.975 | 8.389 | 12.870 | 17.890 |

Table 2 reports the results of Pearson correlation analysis for all variables in this study. Except for the slightly higher correlation coefficient (0.517) between Size and Lev, the correlation coefficients between all variables were less than 0.5, indicating no serious multicollinearity between the variables. In addition, the correlation coefficient between the implementation of the "10-point Soil Plan" (Treat × Post) policy and the SGR of firms was 0.031 and was significantly positive at the 1% level, which tentatively confirms the research hypothesis of this paper.

**Table 2.** Correlation matrix.

| Variables | SGR | Treat × Post | Size | Lev | Age | Soe | First | ID | MKT |
|---|---|---|---|---|---|---|---|---|---|
| SGR | 1 | | | | | | | | |
| Treat × Post | 0.031 *** | 1 | | | | | | | |
| Size | 0.137 *** | 0.023 *** | 1 | | | | | | |
| Lev | −0.124 *** | −0.064 *** | 0.517 *** | 1 | | | | | |
| Age | −0.013 * | 0.099 *** | 0.135 *** | 0.148 *** | 1 | | | | |
| Soe | −0.017 ** | −0.078 *** | 0.336 *** | 0.270 *** | 0.205 *** | 1 | | | |
| First | 0.106 *** | −0.030 *** | 0.218 *** | 0.075 *** | −0.072 *** | 0.242 *** | 1 | | |
| ID | −0.009 | −0.038 *** | −0.005 | −0.011 | −0.029 *** | −0.063 *** | 0.036 *** | 1 | |
| MKT | 0.001 | 0.143 *** | 0.017 ** | −0.029 *** | 0.202 *** | −0.097 *** | −0.044 *** | 0.006 | 1 |

Note. ***, **, and * denote significance at the 1%, 5%, and 10% levels, respectively.

### 4.2. Results of Baseline Regression Analysis

The parallel trend assumption is the basic premise of the DID, which requires that the treatment and control groups maintain the same or similar trends before the implementation of the "10-point Soil Plan" policy. Drawing on Beck et al. [49] and Zeng et al. [50], this study adopted the dynamic DID method to evaluate parallel trends by setting the year

of the beginning of the "10-point Soil Plan" policy as the current dummy variable and setting the dummy variables for several years before (pre) and several years after (post) the policy, respectively.

The dummy variables were set as current dummy variables for the starting year of the "10-point Soil Plan" policy, and the dummy variables were set for several years before (pre) and after (post) the policy, respectively, with pre_3 for 2013, pre_2 for 2014, and so on. After excluding pre_1, these annual dummy variables were incorporated into the model (2) for regression. Figure 2 presents the results of the parallel trend test. The results of regression before 2016 (current) were mostly close to zero, and the 95% confidence interval also contains zero, indicating that the coefficient was not significant, which satisfies the assumption that the treatment and control groups exhibit the same trend before the policy implementation. In contrast, the estimated coefficients were significant and increased gradually from the results three years after the implementation of the "10-point Soil Plan" policy, indicating that the "10-point Soil Plan" policy was effective and significantly promoted corporate sustainability.

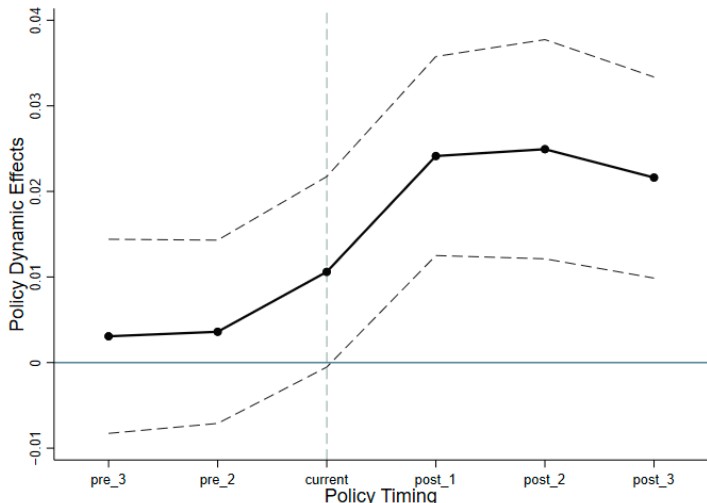

**Figure 2.** Parallel trend test of the "10-point Soil Plan". Note: The solid line indicates the estimated coefficients of the dummy variables associated with the sustainable growth rate of the firm each year. The dashed line represents the 95% confidence interval of the estimated coefficients.

Table 3 reports the impact of the regression of the "10-point Soil Plan" on corporate sustainability. Overall, the implementation of the "10-point Soil Plan" resulted in a positive impact on the sustainable growth rate of firms, i.e., the implementation of the "10-point Soil Plan" was conducive to corporate sustainability. Specifically, column (1) was controlled for the fixed effects of time and city; column (2) was controlled for the fixed effects of time and firm; and column (3) was controlled for the fixed effects of time, firm, and city. The sustainable growth rate of firms in key regulated industries increased by about 2% after the implementation of the "10-point Soil Plan". The hypothesis of this study was validated.

**Table 3.** The impact of the "10-point Soil Plan" on corporate sustainability.

| Variables | SGR | | |
|---|---|---|---|
| | (1) | (2) | (3) |
| Treat × Post | 0.009 *** | 0.020 *** | 0.021 *** |
| | (3.500) | (4.938) | (4.988) |
| Size | 0.023 *** | 0.050 *** | 0.049 *** |
| | (19.080) | (13.914) | (13.308) |
| Lev | −0.130 *** | −0.278 *** | −0.275 *** |
| | (−15.569) | (−17.347) | (−16.789) |

**Table 3.** *Cont.*

| Variables | SGR | | |
|---|---|---|---|
| | (1) | (2) | (3) |
| Treat × Post | 0.009 *** | 0.020 *** | 0.021 *** |
| Age | 0.010 *** | 0.046 * | 0.049 ** |
| | (2.856) | (1.832) | (1.962) |
| Soe | −0.014 *** | −0.025 *** | −0.024 ** |
| | (−5.521) | (−2.630) | (−2.445) |
| First | 0.057 *** | 0.113 *** | 0.109 *** |
| | (7.550) | (5.261) | (5.041) |
| ID | −0.019 | −0.010 | −0.010 |
| | (−0.952) | (−0.341) | (−0.326) |
| MKT | 0.003 | 0.001 | 0.001 |
| | (1.289) | (0.298) | (0.516) |
| _cons | −0.499 *** | −1.129 *** | −1.129 *** |
| | (−11.348) | (−10.273) | (−9.728) |
| Year | Y | Y | Y |
| Firm | N | Y | Y |
| City | Y | N | Y |
| N | 18 687 | 18 687 | 18 687 |
| $R^2$ | 0.120 | 0.434 | 0.440 |

Notes. The results use firm-level cluster robust standard error. The t-statistics are indicated in parentheses. ***, **, and * denote significance at the 1%, 5%, and 10% levels, respectively.

### 4.3. Tests of Robustness

#### 4.3.1. Placebo Test

In order to exclude potential omitted variables, this study conducted a placebo test by randomly generating the "10-point Soil Plan" for key regulatory firms. Since the "fake" treatment group was randomly generated, the implementation variables of the "10-point Soil Plan" policy should not have a significant impact on corporate sustainability, i.e., the regression coefficients of the "fake" treatment variables should be around zero. Accordingly, the above stochastic process was repeated 500 times for model estimation in this study.

Figure 3 presents the kernel density plots of the estimated coefficients of the 500 model experiments. It was found that the mean values of the estimated coefficients were close to zero, and most of the $p$-values were above 0.1, while the true estimated coefficients of this study (0.021) were within the range of small probability events in the kernel density plots. This indicates that the contribution of the "10-point Soil Plan" to the corporate sustainability does not depend on unobserved chance factors, and thus the findings of this paper were reliable and robust.

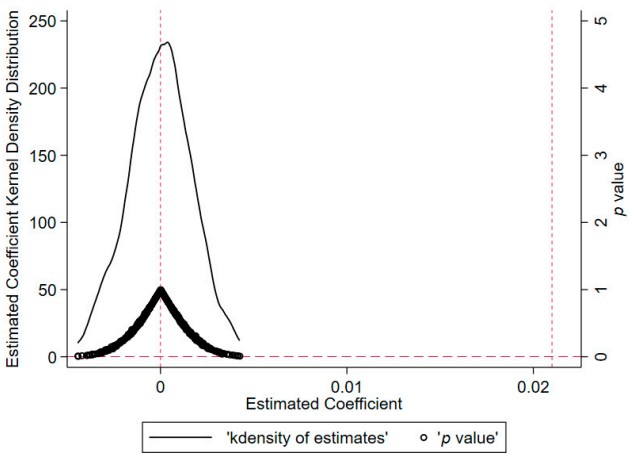

**Figure 3.** Placebo test results.

#### 4.3.2. Propensity Score Matching (PSM)

The "10-point Soil Plan" was not completely random in determining the priority of industries for regulation, and these industries may carry specific factors that affect the sustainability of firms, i.e., intrinsic limitation due to selection bias. For this reason, this study used the PSM method to alleviate the bias. The matching method used in this study was 1:1 based on nearest neighbor matching, and the covariates were all continuous in the control variables. Column (1) of Table 4 reports the results of the matched sample regression with a coefficient $\beta$ of 0.023 for the DID term (Treat × Post) and significant at the 1% level. The results of the PSM-DID estimation were consistent with the previously reported results of the estimation. The estimated coefficients were not significantly different, further indicating the robustness of the study results.

**Table 4.** Results of robustness tests.

| Variables | PSM | Alternative Variable Measures | | | Exclusion of Other Policies | |
|---|---|---|---|---|---|---|
| | (1) | (2) | (3) | (4) | (5) | (6) |
| | SGR | Tobin Q | ΔROA | ΔROS | SGR | |
| Treat × Post | 0.023 *** | 0.101 ** | 0.377 ** | 0.488 *** | 0.020 *** | 0.020 *** |
| | (4.032) | (2.478) | (2.502) | (3.630) | (4.815) | (4.684) |
| Size | 0.042 *** | −0.522 *** | 1.031 *** | 0.948 *** | 0.049 *** | 0.045 *** |
| | (6.902) | (−10.722) | (7.727) | (8.171) | (13.352) | (9.838) |
| Lev | −0.284 *** | 0.073 | −5.849 *** | −5.055 *** | −0.275 *** | −0.225 *** |
| | (−10.855) | (0.414) | (−10.542) | (−10.191) | (−16.808) | (−11.545) |
| Age | 0.073 * | 0.483 * | 1.437 | 0.300 | 0.047 * | 0.024 |
| | (1.960) | (1.784) | (1.596) | (0.366) | (1.888) | (0.758) |
| Soe | −0.004 | −0.282 *** | −0.590 | −0.565 * | −0.023 ** | −0.024 |
| | (−0.232) | (−4.015) | (−1.608) | (−1.903) | (−2.436) | (−1.597) |
| First | 0.074 ** | −0.640 *** | 2.470 *** | 1.924 *** | 0.108 *** | 0.106 *** |
| | (2.365) | (−2.979) | (3.312) | (2.967) | (4.994) | (4.037) |
| ID | −0.028 | 0.289 | −1.797 | −1.208 | −0.009 | −0.021 |
| | (−0.558) | (0.936) | (−1.509) | (−1.172) | (−0.303) | (−0.603) |
| MKT | 0.008 * | 0.041 | −0.005 | 0.138 | 0.002 | 0.001 |
| | (1.864) | (1.248) | (−0.039) | (1.357) | (0.605) | (0.420) |
| CEPI × Post | | | | | −0.008 ** | |
| | | | | | (−2.015) | |
| _cons | −1.108 *** | 12.234 *** | −25.276 *** | −22.363 *** | −1.126 *** | −0.979 *** |
| | (−6.098) | (9.239) | (−6.017) | (−6.127) | (−9.707) | (−6.801) |
| Year | Y | Y | Y | Y | Y | Y |
| Firm | Y | Y | Y | Y | Y | Y |
| City | Y | Y | Y | Y | Y | Y |
| N | 8 706 | 18 687 | 18 687 | 18 687 | 18 687 | 12 596 |
| $R^2$ | 0.532 | 0.721 | 0.228 | 0.230 | 0.440 | 0.490 |

Notes. The results use firm-level cluster robust standard error. The t-statistics are indicated in parentheses. ***, **, and * denote significance at the 1%, 5%, and 10% levels, respectively.

#### 4.3.3. Alternative Variable Measures

To avoid the impact of variable measurement bias on the study results, this study attempted to vary the explanatory variable measures for robustness. Apart from the sustainable growth rate, this study used market value (Tobin Q) to determine the change in return on total assets (ΔROA) and the change in profitability on sales (ΔROS) to represent corporate sustainability from the perspective of market performance and financial performance, respectively, following Combs et al. [51] and Fan et al. [52]. The market value reflects the expected future value of the firms, while the change in return on total assets and the change in the profitability of sales ensure the reliability of the indicator and measures the long-term growth of the firms. The results in columns (2)–(4) of Table 4 show that the coefficients $\beta$ of the DID term (Treat × Post) were significant at the 1% level. It can be seen

that even in terms of market performance and financial performance, the "10-point Soil Plan" indeed enhanced corporate sustainability.

4.3.4. Exclusion of Other Policies

1.  Exclusion of the first round of Central Environmental Protection Inspectors. Considering that the year of implementation of the "10-point Soil Plan" was 2016, environmental policies such as the "13th Five-Year Plan" Environmental Impact Assessment Reform Implementation Plan and the first round of Central Environmental Protection Inspectors were implemented during the same period, but the former focused on regulating the same industries as the "10-point Soil Plan". The urban fixed effects were controlled for in the previous regression analyses to better eliminate the potential impact of other environmental policies implemented by administrative districts. The Central Environmental Protection Inspectors were particularly concerned with coal, chemical, electrolytic aluminum, and thermal power industries, which overlap with the key industries regulated by the "10-point Soil Plan" and may interfere with the conclusions of this study. Therefore, this study further controlled for the interaction term (CEPI × Post) between the first round of CEPI and the policy time point (Post) on the basis of model (2). The robust results of regression analysis are shown in column (5) of Table 4.

2.  Exclusion of Soil Pollution Prevention and Control Law. The Soil Pollution Prevention and Control Law (the Soil Law) was officially implemented nationwide on 1 January 2019. The Soil Law fills the gap in China's soil pollution prevention and control legislation and marks the initial completion of China's soil pollution prevention and control system. Considering that the Soil Law is a complete legal system with stronger constraints than the "10-point Soil Plan" and its influence is broader, this study truncated the sample period, eliminated the observations in 2019 and later, and re-ran the regression on the basis of model (2). The results of the regression are shown in column (6) of Table 4. The coefficient $\beta$ of the DID term (Treat × Post) was significant at the 1% level. The study results are still robust.

**5. Analysis of Mechanism**

Although previous empirical results demonstrated that the implementation of the "10-point Soil Plan" significantly contributed to corporate sustainability, it remains to be further verified whether this was achieved through the debt vacating effect, the cash defense effect, or the innovation compensation effect.

*5.1. Analysis of Debt Vacating Effect*

Local governments derive a large part of their revenue from land transaction fees. The "10-point Soil Plan" reduced local government revenue by strengthening the supervision of the land transfer process, limiting the expansion of local debt, and thus vacating credit for local firms and promoting their sustainable development. To determine whether the "10-point Soil Plan" reduced land revenue, this study followed Zhong et al. [53] and characterized land finance (Land) as the natural logarithm of per capita revenue from land transaction fees. Column (1) of Table 5 presents the results of regression, and the coefficient of "10-point Soil Plan" on land finance was −0.088, which was significant at the 1% level, indicating that "10-point Soil Plan" significantly reduced land finance revenue, freed up credit resources for local firms, and promoted corporate sustainability.

**Table 5.** Results of regression analysis of mechanism.

| Variables | Land Finance (1) Land | Capital Expenditures (2) Capex | Green Innovation (3) lnGreen |
|---|---|---|---|
| Treat × Post | −0.088 *** | −0.076 ** | −0.027 |
| | (−3.661) | (−1.973) | (−0.615) |
| Size | 0.029 | 1.169 *** | 0.322 *** |
| | (1.392) | (27.732) | (7.127) |
| Lev | 0.173 *** | −0.467 *** | −0.279 * |
| | (2.726) | (−3.436) | (−1.931) |
| Age | 0.097 | −0.408 * | −0.527 * |
| | (0.498) | (−1.791) | (−1.779) |
| Soe | 0.064 | −0.195 *** | −0.025 |
| | (0.932) | (−2.595) | (−0.262) |
| First | −0.098 | 0.451 ** | −0.058 |
| | (−0.814) | (2.041) | (−0.242) |
| ID | 0.073 | −0.128 | −0.082 |
| | (0.431) | (−0.466) | (−0.261) |
| MKT | −0.024 | 0.010 | 0.010 |
| | (−1.252) | (0.365) | (0.251) |
| _cons | 7.573 *** | −6.248 *** | −3.841 *** |
| | (10.122) | (−5.388) | (−2.857) |
| Year | Y | Y | Y |
| Firm | Y | Y | Y |
| City | Y | Y | Y |
| N | 7 867 | 18 677 | 7 332 |
| R$^2$ | 0.883 | 0.859 | 0.796 |

Notes. The results use firm-level cluster robust standard error. The t-statistics are provided in parentheses. ***, **, and * denote significance at the 1%, 5%, and 10% levels, respectively.

### 5.2. Determination of Cash Defense Effect

The implementation of the "10-point Soil Plan" resulted in stricter access to land, which partially limited the capital expenditures of firms, thus improving the level of cash holdings of firms and thus promoting their sustainable development. In order to determine whether the "10-point Soil Plan" reduced the firms' capital expenditure, this study used the natural logarithm of total capital expenditures to measure capital expenditures (Capex). Capex refers to the net expenditures incurred for the acquisition and construction of assets with a useful life of more than one fiscal year (e.g., fixed assets, intangible assets, and other long-lived assets), following the approach of King et al. [54]. The results of the regression are reported in column (2) of Table 5. The coefficient of "10-point Soil Plan" on corporate capital expenditures was −0.076 and significant at the 5% level, indicating that the "10-point Soil Plan" promoted corporate sustainability by reducing corporate capital expenditures, which was consistent with the conclusions of this study.

### 5.3. Analysis of Innovation Compensation Effect

According to the Porter hypothesis, environmental regulatory pressures can promote green innovation in firms, allowing them to be compensated for innovation and achieve long-term growth [38,55]. Referring to the research of Ma et al. [56], green patents are used as a proxy variable for green innovation (lnGreen), which is taken by the natural log of the patent number plus 1. The results of the regression are reported in column (3) of Table 5, and the coefficient of the "10-point Soil Plan" on the green innovation of firms was −0.027, but it is not significant, which is contrary to the study of Du et al. [57]. For this, we argue that it may be related to sample selection and the mechanism test method. This indicates that we cannot consider the "10-point Soil Plan" as a general environmental regulation policy and analyze the internal rationale of the innovation compensation effect to enhance corporate sustainability. In this regard, this paper conducted the following analysis.

First, the role of command-and-control environmental regulations on firm innovation is constrained by the stringency of local government environmental regulations [58], and the implementation of the "10-point Soil Plan" is no exception. The "10-point Soil Plan" places particular emphasis on the monitoring of soil environmental quality and the control of soil pollution risks. This "top-down" environmental regulation relies on local governments to promote implementation. However, unclear responsibilities have been one of the main challenges in soil pollution management in the past [8]. While the role of local governments is important, firms are responsible for soil pollution and control. The only indicator used in "10-point Soil Plan" was the "safe utilization rate of contaminated land", which was also an assessment target for local governments without a deterrent effect on firms; hence, it is difficult for firms to be compelled by the "10-point Soil Plan" to implement green innovation spontaneously.

Second, the process of environmental regulation to promote green innovation, which in turn drives technological progress and ultimately productivity growth, is not a one-day process [59]. The corporate strategies under the "10-point Soil Plan" can be divided into "source-control" and "end-treatment". If firms respond to the "10-point Soil Plan" by making temporary investments such as the direct purchase of production equipment, environmental regulations will not be able to substantially promote the development of innovative production methods [60]. Finally, it should not be overlooked that China's soil pollution remediation technology is still relatively backward as a whole. To ensure compensation for innovation, the government needs to conduct independent technical research and introduce advanced foreign technology concurrently, to build a prevention-oriented system for soil environmental management.

In summary, the "10-point Soil Plan" promoted corporate sustainability mainly via debt "vacation" and cash defense but failed to achieve the innovation compensation effect. Accordingly, the "10-point Soil Plan" prompted local governments to take administrative measures, such as reducing land finance and strict land access. These local government actions eased the financing constraints of firms and increased their cash holdings. However, these were passively accepted by firms. Conversely, the innovation compensation effect was more demanding on firm autonomy, requiring firms to take the initiative to assume responsibility for soil pollution prevention and control. The innovation compensation effect has not been achieved, indicating that the "10-point Soil Plan" has yet to directly change the environmental behavior of firms, further indicating that the current positive effect of the "10-point Soil Plan" on micro-firms mainly relies on administrative intervention by the local governments, and the awareness of firms to assume responsibility has yet to be stimulated.

## 6. Heterogeneity Analysis

Differences exist between firms in terms of region and resource endowment, and whether these differences cause heterogeneity in the implementation of the "10-point Soil Plan" policy needs further investigation.

### 6.1. Nature of Property Rights

This study divided the sample into state-owned enterprises (SOEs) and non-state-owned enterprises (non-SOEs) according to the nature of property rights and tested the impact of the "10-point Soil Plan" on corporate sustainability. The coefficient of the DID term (Treat × Post) in column (1) of Table 6 for the SOE sample is 0.040, which passes the significance test at the 1% level, while the coefficient of the DID term (Treat × Post) in column (2) of Table 6 for the non-SOEs sample is 0.009, which passes the significance test at 10% level. The *p*-value of the difference between the two groups is close to 0, indicating that the implementation of the "10-point Soil Plan" policy has a highly significant impact on the sustainability of SOEs compared with non-SOEs.

Two potential factors contribute to this effect: First, the "10-point Soil Plan" reflects the high importance that the central government attaches to soil pollution prevention and control, and the central government's awareness is often implemented by SOEs, which are

under great pressure to fulfill their political mission [61]. The original text of the "10-point Soil Plan" also emphasizes that SOEs should take the lead with a stronger sense of social responsibility. They are monitored by local governments and are provided with stronger incentives to maintain good performance. Second, China has a highly centralized financial system dominated by state-owned commercial banks, and there is an institutional order of dominance in the allocation of credit [62]. Therefore, the advantages of SOEs in terms of financial support and financing capacity [63] ensure debt vacating and cash defense effects, thus achieving corporate sustainability.

**Table 6.** Results of regression analysis of heterogeneity.

| Variables | SOEs (1) | Non-SOEs (2) | Large Firms (3) | Small Firms (4) | High ER Areas (5) | Low ER Areas (6) |
|---|---|---|---|---|---|---|
| | SGR | | SGR | | SGR | |
| Treat × Post | 0.040 *** | 0.009 * | 0.031 *** | 0.008 | 0.027 *** | 0.008 |
| | (5.400) | (1.828) | (5.204) | (1.317) | (4.941) | (1.353) |
| Size | 0.040 *** | 0.057 *** | | | 0.049 *** | 0.051 *** |
| | (5.889) | (12.163) | | | (10.693) | (6.793) |
| Lev | −0.291 *** | −0.243 *** | −0.265 *** | −0.230 *** | −0.284 *** | −0.260 *** |
| | (−9.753) | (−12.216) | (−10.148) | (−9.715) | (−14.091) | (−8.686) |
| Age | 0.118 *** | 0.053 | 0.061 | 0.023 | 0.076 ** | 0.028 |
| | (2.727) | (1.613) | (1.559) | (0.561) | (2.226) | (0.700) |
| Soe | | | −0.013 | −0.021 | −0.021 | −0.040 *** |
| | | | (−0.900) | (−1.367) | (−1.524) | (−2.803) |
| First | 0.063 * | 0.130 *** | 0.102 *** | 0.137 *** | 0.134 *** | 0.089 ** |
| | (1.731) | (4.529) | (3.537) | (3.813) | (4.967) | (2.135) |
| ID | −0.067 | 0.017 | 0.004 | −0.085 * | −0.051 | 0.047 |
| | (−1.616) | (0.425) | (0.098) | (−1.895) | (−1.278) | (1.002) |
| MKT | −0.003 | 0.004 | 0.004 | −0.003 | 0.002 | −0.008 |
| | (−0.729) | (1.149) | (1.101) | (−0.713) | (0.632) | (−1.446) |
| _cons | −1.034 *** | −1.382 *** | −0.082 | 0.077 | −1.207 *** | −0.999 *** |
| | (−5.071) | (−9.175) | (−0.657) | (0.575) | (−8.396) | (−4.525) |
| Year | Y | Y | Y | Y | Y | Y |
| Firm | Y | Y | Y | Y | Y | Y |
| City | Y | Y | Y | Y | Y | Y |
| p-value | 0.000 *** | | 0.000 *** | | 0.000 *** | |
| N | 7 064 | 11 623 | 9 311 | 9 376 | 11 612 | 7 075 |
| R² | 0.481 | 0.457 | 0.472 | 0.477 | 0.517 | 0.550 |

Notes. The results use firm-level cluster robust standard error. The t-statistics are provided in parentheses. ***, **, and * denote significance at the 1%, 5%, and 10% levels, respectively. Empirical *p*-values were used to test the significance of differences in Treat × Post coefficients between groups, obtained by Bootstrapping 1000 times.

### 6.2. Firm Size

In order to analyze the relationship between the "10-point Soil Plan" and the corporate sustainability of different sizes, this study divided the sample into large and small firms, using the median size of firms as the criterion. The results of group regression are shown in columns (3) and (4) of Table 6. The coefficient of the DID term (Treat × Post) in column (3) for the large firms is 0.031, which passes the significance test at the 1% level, while the coefficient of the DID term (Treat × Post) in column (4) for small firms is 0.008, which does not pass the significance test. The *p*-value of the difference between the two groups is close to 0, indicating that the implementation of the "10-point Soil Plan" policy has a highly significant impact on the sustainability of large firms compared with small firms.

Large firms have more financial advantages and better financing capacity than small firms [64] and are less prone to financial distress. Further, large firms have greater resources to engage in sustainable developmental activities [65] and sustain their long-term growth. Thus, large firms exhibit more pronounced policy-enhancing effects of environmental regulations.

*6.3. Local Environmental Regulation Intensity*

As confirmed earlier in this study, the innovation compensation effect of the "10-point Soil Plan" depends on a reasonable intensity of environmental regulation. The pressures of environmental regulation can motivate firms to actively participate in environmental management, as well as local governments, where the degree of environmental regulation in the framework of local government actions is an important component of effective policy transmission [66]. Therefore, this study suggests possible differences in the impact of the "10-point Soil Plan" on areas with high and low environmental regulation intensity.

Accordingly, we measured the intensity of regional environmental regulation from the perspective of investment in pollution control, borrowing from Wang et al. [67]. The ratio of total investment in pollution control to industrial value added in each province (ER) was directly correlated with the intensity of environmental regulation. Based on the median value of environmental regulation intensity, the sample was divided into high and low intensity areas of environmental regulation. The results of group regression are presented in columns (5) and (6) of Table 6. The coefficient of the DID term (Treat $\times$ Post) is 0.027 and is significant at the 1% level for areas with high environmental regulation intensity in column (5). The coefficient of the DID term (Treat $\times$ Post) is 0.008 for areas with low environmental regulation intensity in column (6), which does not pass the significance test. The *p*-values of the differences between the groups are all significant at the 1% level. The above empirical results show that the promotion of the "10-point Soil Plan" on the sustainability of firms is more significant in areas with high environmental regulation intensity, which further echoes the effect of innovation compensation effect, suggesting the need for regulatory pressure to ensure active response by firms.

**7. Conclusions**

This study investigated the impact of the "10-point Soil Plan" on corporate sustainability, deconstructed its internal mechanism, and further analyzed the heterogeneity of the policy effects. The findings suggest that the implementation of the "10-point Soil Plan" has significantly contributed to the sustainability of firms in key regulated industries. This sustainability is attributed to the improved financial status of companies following administrative intervention by local governments, rather than environmental regulation based on the "10-point Soil Plan". The anticipated innovation compensation effect of the "10-point Soil Plan" has yet to be realized, which is contrary to the study of Du et al. [57]. It may be related to the regulatory intensity of the policy, the temporary environmental response of firms, and the overall low level of soil pollution remediation technology. Further analysis reveals that the sustainability of firms under the "10-point Soil Plan" is more significant in state-owned enterprises, large firms, and regions with high environmental regulation intensity.

Although the "10-point Soil Plan" is a relatively systematic regulation in the field of soil pollution prevention and control, urgent improvements are still needed for the prevention and control of soil pollution in China. The study findings provide the following policy insights and recommendations:

First, from the perspective of an optimization of the "10-point Soil Plan", the environmental protection departments need to strengthen the identification of pollution responsibility for firms. Thus, the policy has a deterrent effect on firms and ensures their positive environmental response and behavior.

Second, from the perspective of local government enforcement, a reasonable intensity of environmental regulation is required to ensure that the intensity of environmental regulation exceeds the threshold of environmental supervision. In addition, local governments can adjust their strategies to achieve the phased objectives of soil pollution prevention and control via long-term efforts by firms.

Third, from an industrial perspective, companies should strengthen their strategic management to better cope with environmental policy shocks. Heavy polluters should abandon short-sightedness when making strategic decisions and seek corporate sustain-

ability, such as introducing advanced foreign pollution control technologies and innovative green technology to ensure clean production.

Fourth, from a social perspective, local governments should encourage commercial banks to actively participate in environmental governance, dynamically adjust credit resources, and increase credit disbursement to small, non-state heavy polluters that have difficulties with technological innovation. Public participation can be increasingly utilized for the assessment and investigation of polluted land, and supervision of local governments and firms for the inclusion and restoration of polluted land.

The study complements the systematic study of soil environmental planning and helps China integrate soil environmental planning with water and air environmental planning to build a comprehensive pollution prevention system, but it may also have the following research limitations. First, although the "10-point Soil Plan" does promote corporate sustainable development, corporate sustainable development is a more comprehensive concept, and the empirical process contains more confounding factors. Second, the study does not make a further distinction between soil pollution industries, such as considering heavy metal industries and non-heavy metal industries, etc. How to further improve the above issues will be the direction of future research.

**Author Contributions:** Conceptualization, Q.Z. and P.Z.; investigation, Q.T. and P.Z.; methodology, Q.T., H.Z. and Y.-E.L.; supervision, Q.Z., H.Z. and P.Z.; visualization, Q.Z. and Y.-E.L.; writing—original draft, Q.T. and P.Z.; writing—review and editing, Q.T., H.Z. and Y.-E.L. All authors have read and agreed to the published version of the manuscript.

**Funding:** This research was funded by the National Natural Science Foundation of China, grant number [71904208], the Youth Project for Nature Science Foundation of Hunan Province (China), grant number [2021JJ40796], the Key Project of Social Science Foundation of Hunan Province (China), grant number [22ZDB042], and the Central South University Innovation-Driven Research Programme, grant number [2023CXQD035].

**Institutional Review Board Statement:** Not applicable.

**Informed Consent Statement:** Not applicable.

**Data Availability Statement:** Not applicable.

**Conflicts of Interest:** The authors declare no conflict of interest.

## Appendix A

**Table A1.** Variables.

| Variable | Definition |
|---|---|
| SGR | Calculated using Van Horn's SGM model, see Equation (1). |
| Treat × Post | Treat is 1 when the firm belongs to the industries regulated by the "10-point Soil Plan"; otherwise, it is zero. Post is 1 for 2016 and subsequent years; otherwise, it is zero. |
| Size | Natural logarithm of the firm's total assets at the end of the year. |
| Lev | Total liabilities divided by total assets. |
| Age | Natural logarithm of the firm's age of establishment. |
| Soe | State-owned enterprises are assigned a value of 1, while non-state-owned enterprises are assigned zero. |
| First | Number of shares held by the first largest shareholder/all shares. |
| ID | Number of independent directors/total number of board of directors. |
| MKT | Refers to "China Marketization Index Report by Province (2021)"; missing years are filled in by manual calculation. |
| $\gamma_t$ | The aim is to control for characteristics that do not vary with the individual at different times. |
| $\gamma_i$ | The aim is to eliminate heterogeneity between firms. |
| $\gamma_{city}$ | The aim is to eliminate heterogeneity between cities. |

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
