# Peer review of "Does Soil Pollution Prevention and Control Promote Corporate Sustainable Development? A Quasi-Natural Experiment of “10-Point Soil Plan” in China"

_sustainability, doi:10.3390/su15054598_

Round 1

Reviewer 1 Report

Addition a separate section for conclusions.

1.What is the main question addressed by the research?

The main question that the research addressed is whether it is possible to prevent and control soil pollution, especially with heavy metals.

2. Do you consider the topic original or relevant in the field? Does it address a specific gap in the field?

It is relevant in the field and it addresses a specific gap in the field.

3. What does it add to the subject area compared with other published material?

It adds a tested method for preventing and controlling soil pollution with the contribution of corporates that represent one of the sources of pollution for a sustainable environment and a sustainable economy.

4. What specific improvements should the authors consider regarding the methodology?

What further controls should be considered? The methodology used needs to be presented in the form of specific steps and to specify the nature of the data required to test each step.

5. Are the conclusions consistent with the evidence and arguments presented and do they address the main question posed?

The paper did not include a conclusions section and I have already made this observation in my comment to the authors. After adding this section, the question can be answered.

6. Are the references appropriate?

Yes

7. Please include any additional comments on the tables and figures.

I have not any additional comments on the tables and figures.

Reviewer 2 Report

Thank you for the opportunity to review this manuscript. As a researcher in this field myself, I very much enjoyed reading it. This paper examines the effect and mechanism of the “10-point Soil Plan” on corporate sustainable development in China. While the article is informative and of potential interest among some readers, I found it difficult to justify the rationale and motivation for conducting the research. Below I describe my concerns in greater detail. I hope that my comments are viewed as constructive feedback that will help you clarify and enhance your research quality. My comments are as follows:

Title;- Sound !

Keywords;- the authors could remove the quotation marks from the keywords if not necessary.

Abstract;-

-        Sound! However, the authors could add on the finds rather than just describing the relationship. The authors could provide more detail about what the findings mean and what implications they have for the field. This could help to further engage readers and provide a deeper understanding of the research.

Introduction;- Apart from the major concern discussed above, the authors need to consider the followings;

-        The following statement is not clear “ignoring the fact that the “10-point Soil Plan” also has a preventive role.”

-        I agree with the observation made in the article that the authors have provided a detailed background on the 10-point Soil Plan in China, but have failed to adequately address the role of firms in drawing the research rationale. The statement "few studies empirically tested the policy effects of the 10-point Soil Plan" without considering the main source of soil pollution prevention and control (firms) is not sufficient justification for the study. I suggest the author to enhance the motivation of the study.

Literature review;-

- The authors' lack of a clear theoretical justification for the relationship being investigated is a significant limitation of the study. A solid theoretical framework provides a roadmap for the research, clarifying the purpose and importance of the investigation, and helps guide the analysis and interpretation of the results. The absence of a clear theoretical foundation makes it difficult for the reader to understand the relevance of the study, and to assess the validity of the findings. In future research, it would be useful for the authors to provide a more comprehensive theoretical justification for the relationship they are exploring. This could involve a review of the relevant literature, a clear articulation of the research questions and hypotheses, and a discussion of the implications of the findings for theory and practice.

- in the following statement “Based on the agency theory of cash holdings, excessive cash holdings 212 are prone to generate agency conflicts and damage the market value of the firms” refer to the following study (Shariah auditing: analyzing the past to prepare for the future auditing)

Methodology;-

-        Provide reference for the measurement of the research variables. “including firm size (Size), leverage (Lev), firm age (Age), nature of ownership (Soe), 309 ownership concentration (First), board independence (ID), dual position (Dual), and re- 310 gional marketability index (MKT). The definitions and measures of all variables are sum- 311 marized” measurement of these variables are used in the following studies “Financial Inclusion and the Performance of Banking Sector in Palestine” and “ESG performance in the time of COVID-19 pandemic: cross-country evidence”

Results;

It is important for the authors to present their findings in a clear and compelling manner, making a strong argument for their conclusions. The findings should be discussed in a way that highlights their significance and provides a clear explanation of how they support the authors' overall argument. The authors should also critically examine the limitations of the study and provide context for how the findings fit into the broader landscape of research on the topic. By taking a more argumentative approach, the authors can provide a stronger and more impactful contribution to the field.

'

Conclusion;

-        The heading “7. Discussion” should be changed to “7. Conclusion”

-        I do appreciate the author effort in presenting the research implications. However, I found that the four points highlighted do not provide enough reasoning or evidence to support why these recommendations should be followed/taken. It would be beneficial for the authors to elaborate on the implications and consequences of not following these recommendations and provide evidence or examples to support their stance."

-        The authors need to discuss the limitation of the study that could be use in future research. In general, identifying the limitations of a study can provide direction for future research and help improve upon the current findings. It is important for researchers to acknowledge the limitations of their study and suggest areas for future research that could address these limitations. This can lead to a better understanding of the topic and contribute to the overall advancement of the field.

There are a few of typos/grammatically incorrect sentences in the manuscript that need to be remedied. Please note that I provided just single examples below. The paper needs to be checked thoroughly to meet the standard required in this journal.
1     
There is a lack of smooth transition from one paragraph to another.

2     In-text reference issue. Authors need to use the correct intext citations as required by the journal:  “Overall, most scholars are not optimistic about the “10-point Soil Plan”. Hou and Li 69 (2017) [10] point out that soil pollution prevention and control have externalities and spill- 70 over effects ranging from greenhouse gas emissions to social justice. According to Li et al. 71 (2019) [11]” another example “Al-Shaer and Hussainey (2022) and Lu et al. (2022) [39,40],” which suppose to be as follow “Al-Shaer and Hussainey [390 and Lu et al. [40],”

3     Abbreviation should be defined first time used and consistently use it; “ST, *ST and PT categories;”. Also, abbreviation should be defined only at first time used; “e.g., firm size (Size)”

4     Consistency in the capitalization; “3.2.2. Independent variable” and “4.3.4. Exclusion of other policie”

I believe your paper has merit and I am certain that it can be significantly improved. Once these comments addressed the paper will make an important contribution to the literature on this subject, Overall, I am delighted to read this manuscript. I hope that my comments and questions will provide the authors with some guidance to improve their manuscript.

Best of luck to you!

Reviewer 3 Report

Overall, the paper is complex from both theoretical background and model analysis perspective, being grounded on recent and relevant papers in the field of corporate sustainability, especially from Asia-Pacific Region. However, I consider there is still place for some improvements before the paper is suitable for publication. Comments on different issues which should be improved are provided below. I trust there is a potential for authors to revise and improve the manuscript.

1. The introduction is well set-up, presenting the main gaps in existing research, thus justifying why this study is needed.

However, I believe that the paragraph describing paper’s goal aim (line 87-90) should be a little bit expanded. The authors state that “this study developed a Difference-in-Differences (DID) model to empirically test the micro effects of the “10-point Soil Plan” on firms, and analyze the intrinsic mechanisms and heterogeneity of the effects in the context of the interactive behaviors of local governments and firms.”, but what is a the existing gap? The authors only mentioned that “few studies empirically tested the policy effects 81 of the “10-point Soil Plan”, without considering the fact that firms are the main source of 82 soil pollution prevention and control” (line 81-82) … Are there such studies or not … if yes, than what is the added-value?

It would be useful for the readers to know more clear from the introduction. There is just one paragraph in this respect (line 72-77).  Thus, I have some suggestions for the authors:

- They should focus more on “What does this study do to address the identified gaps?” and to try to underline their effort.

- A short reference to the methodological issues it would be recommended. Does this paper use the same methods as usual? … if not, as I presume, the authors should better underline this issue.

- Clear emphasises of the main findings should be of readers’ interest from the beginning. Even though the results are mentioned (line 90-98) these are presented closely linked with the method used.

- Even though authors detailed at the end of the paper some policy insights and recommendations, I recommend to briefly mention this issue in introduction as well … Whom is this paper useful for?

2. The theoretical background and research hypothesis is based on recent studies in the researched topic, thus being well grounded and up to date. However, I found it difficult to identify the theories the authors relied on to underpin the research hypothesis (If is just one I think it should not be numbered). The authors briefly mentioned two theories: agency theory (line 212) and trade-off theory (line 214).

3. Besides, the authors divided the theoretical background in 3 subsection focused on “Debt “vacating” effect”, “Cash “defense” effect” and “nnovation compensation effect” (2.2.1 to 2.2.3), the three pillars of the proposed model. The results are also analysed separately for the three effects tested (subsections 5.1 to 5.3). This context I believe that it would be suitable to formulate three sub-hypotheses … in this way it would be a clear connection between the theoretical background and the findings of the study

4. The Analysis of Mechanism (section 5) and the Heterogeneity Analysis (section 6) are in-depth presented and explained from the statistical perspective, but the author do not make any reference to prior studies (Have they reached similar or opposite results?).

6. In the same vein as the previous recommendation as suggest that the authors should present their findings (subsection 7 “Discussion”) in close connection with prior evidence, thus emphasizing similarities (if any) or pointing out new and original results.

7. As regards the structure of the paper, I believe that the paper either should end with a separate section called “6. Conclusions” (embedding the implications, limitations and future suggestions … I didn’t find any of these mentioned).

Round 2

Reviewer 2 Report

Thank you for the opportunity to review this manuscript again.

I have reviewed your manuscript titled "Does soil pollution prevention and control promote corporate sustainable development? A quasi-natural experiment of “10-point Soil Plan” in China" and have some feedback that I believe will help to strengthen your work.

Firstly, I suggest that you further support your findings with relevant literature and discuss them in an argumentative manner. While you briefly mention some previous studies in the introduction, there is a lack of comprehensive review of the literature related to the impact of environmental regulations on corporate sustainability. Adding more relevant literature to support your findings will provide a stronger foundation for your research.

Secondly, I suggest revising the manuscript language. For example, in the following statement "This study complements the systematic study of soil environmental planning and helps China integrate soil environmental planning...." it is not clear which study you are referring to. Please clarify this and review the manuscript for similar issues throughout.

Thirdly, I suggest checking and correcting the reference list as it is not as per the journal requirements. Please ensure that all references are properly cited and formatted according to the journal guidelines.

Fourthly, I suggest that you present the limitations of the study at the end of the conclusion section. This will help readers to better understand the scope and generalizability of your findings.

Lastly, there is a concern regarding the selection of the research variables. For example, the factor "Dual" has zero relation to the main construct. Please review the variables selected for the study and ensure that they are relevant and contribute to the research objectives.

Overall, your study provides valuable insights into the impact of the "10-point Soil Plan" on corporate sustainable development in China. Addressing these feedback points will help to strengthen your manuscript and contribute to the literature on environmental regulations and corporate sustainability.

Best of luck!

Reviewer 3 Report

Overall, I appreciate that the paper has improved and the authors are on the right path to a good quality paper, complex from the methodological perspective, grounded on relevant literature and focused on an interesting area  into the economic field, namely sustainable development, which is quite challenging. They consider prior recommendations to improve their work, especially the introduction and the conclusions. 
